# Active Ageing Awareness and Quality of Life among Pre-Elder Malaysian Public Employees

**DOI:** 10.3390/ijerph19159034

**Published:** 2022-07-25

**Authors:** Nor Hana Ahmad Bahuri, Hussein Rizal, Mas Ayu Said, Phyo Kyaw Myint, Tin Tin Su

**Affiliations:** 1Department of Social and Preventive Medicine, Faculty of Medicine, University of Malaya, Kuala Lumpur 50603, Malaysia; drhannadrph@gmail.com (N.H.A.B.); husseinriz@um.edu.my (H.R.); 2Centre for Population Health (CePH), Department of Social and Preventive Medicine, Faculty of Medicine, University of Malaya, Kuala Lumpur 50603, Malaysia; 3Centre for Epidemiology and Evidence-Based Practice, Department of Social and Preventive Medicine, Faculty of Medicine, University of Malaya, Kuala Lumpur 50603, Malaysia; 4Ageing Clinical & Experimental Research Team, Institute of Applied Health Sciences, University of Aberdeen, Aberdeen AB25 2ZD, UK; phyo.myint@abdn.ac.uk; 5South East Asia Community Observatory (SEACO) and Global Public Health, Jeffrey Cheah School of Medicine & Health Sciences, Monash University, Bandar Sunway, Subang Jaya 47500, Malaysia

**Keywords:** active ageing, elderly, low-income, Malaysia, older adults, quality of life

## Abstract

Increasing life expectancy has led to a global rise in late-life diseases. Quality of Life (QOL) is important for healthy life expectancy. The active ageing framework serves as a guide for policymakers to design policies that enhance the QOL of older people. This study aims to determine the association between awareness of active ageing and QOL. The Malay version of the 26-item WHOQOL-BREF questionnaire was utilised along with the 14-item Active Ageing Awareness Questionnaire (AAAQ). A total of 532 participants had a mean (SD) age of 50.2 (5.9), were largely ethnic Malay (96.2%), female (52.8%), and comprised largely of low-income households (65.4%). The median (IQR) AAAQ score was 71.4 (19.1). The hierarchical multiple regression analysis revealed significant positive association between AAAQ and the QOL domains of physical (β = 0.154, *p* < 0.001), psychological (β = 0.196, *p* < 0.001), social relationship (β = 0.175, *p* < 0.001), and environment (β = 0.145, *p* < 0.001) after adjusting for all covariates. Awareness of active ageing was found to have a positive effect on all domains of QOL among pre-elder employees, and thus, we recommend that policies to improve active ageing awareness should be implemented for healthy life expectancy in ageing populations.

## 1. Introduction

Longer human lives have led to a global rise in late-life diseases [1], which has resulted in the endorsement of the active ageing framework championed by the World Health Organisation (WHO) [2]. This framework serves as a guide for policymakers to design policies and programmes that aim to enhance the quality of life (QOL) of older people as they age by optimising opportunities for health, participation, and security. Several studies have measured the level of active ageing and the QOL of older people and have identified the factors associated with these outcomes. The identified factors were (1) Demographics [3,4,5], made up of age, gender, ethnicity, and marital status; (2) Socio-economic status, which includes educational level [5,6], occupation [3,5], income [7], housing [8], financial well-being [9], and owning a vehicle [10]; (3) Health-related factors, including health literacy [11], presence of chronic diseases [4], and health behaviours [12,13] such as tobacco use, unhealthy diet, physical inactivity, harmful use of alcohol, and participation in leisure activities [14]; (4) External factors, ranging from having a pre-retirement plan [15], having a valid driving licence [16], and the living condition of a person’s neighbourhood [17].

QOL is defined as an individual’s perception of their position in their life in the context of the culture and value systems in which they live and in relation to their goals, expectations, standards, and concerns. It is also well acknowledged that the QOL is vulnerable to deterioration due to the development of disease and disability, diminished cognitive capabilities, and devastating life events such as the death of a spouse or being a victim of natural disasters [18]. Examples including addressing financial hardship, limiting the progress of disease, and living in a conducive neighbourhood seem to influence the QOL of older people positively [18]. Thus, to optimise the QOL of older people, it is vital to gain a comprehensive understanding of the current QOL of the adult population and of its determinants. Moreover, as the efficiency of health programmes decreases with advancing age, interventions are more effective in early and middle adulthood [19]. Although late investments in old age are effective, this approach may not be as effective compared to interventions in earlier life, as many people have already developed diseases and are facing complications because of it [20].

By optimising opportunities for health, participation, and security, policymakers will be able to plan early interventions to assist the future older person to age gracefully by enhancing their QOL. Currently, to our knowledge, no study has assessed the awareness of active ageing among the general population, particularly pre-elders. Studies on awareness of various diseases in Malaysia have produced inconsistent findings [21,22,23]. If an individual has no idea about what factors are important for ageing well, they will not make any behaviour changes in order to ensure their QOL. A tool to assess awareness of active ageing has recently been developed to address this problem [24]. As efforts are made in Malaysia to promote and implement active ageing policies and awareness, this framework may provide a window of opportunity through which to scrutinise the potential of future older people to engage in active ageing. This study can be considered among the first to be conducted on this issue in Malaysia. Thus, this cross-sectional study attempts to determine the factors associated with QOL among public employees in Johor, Malaysia. We aimed to test the hypothesis that there is a positive association between awareness of active ageing and QOL.

## 2. Materials and Methods

Ethical approval was obtained from the Ministry of Health, Malaysia (NMRR-16-40-28747), Medical Research and Ethics Committee and Medical Ethics Committee University Malaya Medical Center (MREC ID no: 20161-2037). Participants that volunteered to participate in the study gave written informed consent. Data were collected from 1 April 2017 to 30 November 2017 in the Kluang district of Johor state, Malaysia.

### 2.1. Participants

Simple random sampling was used to select public employees aged between 40 and 60 years. This age range was selected because people in this age group will contribute to the older population when Malaysia becomes an aged nation in 2030, making it the ideal age to prepare for older adult life. Employees from the manager and professional categories, army officers, pregnant women, and employees on sick or long-term leave were excluded from the study. The sample size was calculated based on a 0.05 precision level and a 95% confidence interval with a power of 80%. The association between awareness of active ageing and the four domains of QOL (refer to Section 2.3) was tested using a correlation coefficient. The sample size calculated was 576 after taking into account a 20% non-response rate.

### 2.2. Procedure

A self-administered questionnaire package was given to the participants. Each package contained an information sheet, a consent form and a questionnaire booklet in an envelope and was given to the eligible participants via their department’s representative. They were given one week to complete the questionnaire and return it to their representative. The participants were instructed to place the completed questionnaire in the provided envelope, seal it, and pass it to their department’s representative. The researcher set a date with each representative to visit the respective departments to collect the completed questionnaires.

### 2.3. Dependent Variable

The Malay version of the WHOQOL-BREF questionnaire contained 26 items and was validated among the adult population [25]. The internal consistency was acceptable, with a Cronbach’s alpha of 0.80, 0.64, 0.65, and 0.73 for physical health, psychological, social relationships, and environment domains, respectively. It had two stand-alone questions to measure the individual’s overall perception of QOL and the individual’s overall perception of their health. The remaining 24 items were combined to measure the individual’s perception of their QOL in four domains: physical health (seven items), psychological (six items), social relationships (three items), and environment (eight items). A five-point Likert scale was used to score each item, where the higher the number, the higher the rating. The score for the two stand-alone questions ranged from one to five, where a higher score indicated higher quality of life and health satisfaction. The domain scores ranged from 0 to 100. Initially, the raw score for each domain was obtained. Next, the mean score for each domain was calculated and multiplied by 4 to transform the score into a range between 4 and 20. Then, a second transformation was done to convert the score to a 0–100 scale by multiplying the score in the first transformation by 100 and then dividing it by 16.

### 2.4. Primary Outcome Variable

The newly developed and validated Active Ageing Awareness Questionnaire (AAAQ) consists of two stand-alone questions and 14 items that show satisfactory validity and reliability, with a Cronbach’s alpha of more than 0.7 [24]. The two stand-alone questions were analysed separately. The first question asks whether the participants have heard of the term ‘active ageing’. The second question is an open-ended question, where the participants are asked to give their opinion about factors that may help them to age actively. The responses gathered from question (2) fell outside the scope of this paper and, thus, were not included. The participant’s responses to the 14 items in the AAAQ were scored using a 4-point Likert scale and were summed to form a score ranging from 14 to 56. The score was then converted into a score based on a scale of 0–100.

### 2.5. Secondary Outcome Variable

#### 2.5.1. Demographic

Demographic and socio-demographic details such as age, gender, ethnicity, marital status, education level, and income were recorded. Income salary was categorised into two groups using the Malaysian below 40 (B40) household income as a cut-off point.

#### 2.5.2. Financial Retirement Confidence

The financial retirement confidence adopted by Sabri et al. [26] was also used to assess the confidence level of participants about the financial aspect of their retirement. It represents the financial behaviour of the participants. The tool has eight items and is scored on a four-point Likert scale. The score is the sum of all the items rated by participants, where a higher score indicates a higher confidence level. The final score, which ranges from 8 to 32, is categorised into 3 groups: low (8–16), moderate (17–24), and high (25–32).

#### 2.5.3. Short Form Health Literacy Questionnaire (HL-SF12)

The 12-item Short Form Health Literacy Questionnaire (HL-SF12) was used and has been found to be a valid and reliable tool to assess health literacy [27]. The HL-SF12 employs a four-point Likert scale that assesses the perceived difficulty of performing each item: (1) Inadequate, (2) Problematic, (3) Sufficient, and (4) Excellent. Three health literacy indices: healthcare, disease prevention, and health promotion, are constructed as a General Health Literacy Index.

#### 2.5.4. Other Supporting Factors

Associated factors based on the literature [11,12,13,14,15,16,17] such as the number of chronic diseases, consumption of fruits and vegetables, smoking status, physical activity levels, and external factors such as: attending a pre-retirement course, attending a non-job-related course, owning a vehicle, having a valid driving license, and the number of basic facilities near home were recorded.

### 2.6. Statistical Analysis

#### 2.6.1. Descriptive Analysis

Frequency analysis was carried out to identify missing values and extreme values. Extreme values that were deemed as a case of wrong data entry were deleted and assigned as missing data. The descriptive analysis determined the association between each of the independent variables with the dependent variables. The independent variables are active ageing awareness, demographic factors, financial retirement confidence, health literacy, and other supporting factors listed in Section 2.5.4, while the dependent variables are the four domains of QOL and health satisfaction (refer to Section 2.3).

#### 2.6.2. Inferential Analysis

The univariable analysis was conducted to examine the predetermined significance values of the association between the independent and dependent variables. As the domains of perceived QOL and perceived health status are binomial (poor vs. good), logistic regression was used. The Hosmer–Lemeshow test, Pearson chi-square and classification table were applied to check the model fitness of the final model. Physical health, psychological, social relationships, and environmental domains are continuous variables (ranging from 0–100), and thus, a linear regression was applied.

Multicollinearity and the fitness of the model were examined. The backward stepwise method was applied, and the model assumptions were fulfilled. This approach begins with a full (saturated) model and, at each step, gradually eliminates variables from the regression model to find a reduced model that best explains the data. There were no interactions among independent variables. Multivariable outliers were identified by examining the Mahalanobis distance. The cases with a probability of less than 0.05 were deleted before running the multiple linear regression with the selected independent variables.

Finally, the hypothesis on the association between the awareness of active ageing and the QOL was tested by controlling for other variables via a hierarchical multiple regression.

## 3. Results

### 3.1. Socio-Demographic and Health-Related Characteristics

A total of 700 participants from 46 out of 74 participating public sector departments were combined into the sampling frame and invited for the study. Out of the 700 participants, 532 completed the survey within the time frame. Out of the 532 participants that completed the survey, 213 (40%) had at least one missing value. Little’s MCAR test [28] revealed that the missing values were random (*p* = 0.167) after excluding two variables: ‘total amount of loans’ and ‘monthly loan commitments’. The demographic characteristics (Table 1) showed that the participants had a mean (SD) age of 50.2 (5.9), were largely ethnic Malay (96.2%), were almost proportionally equal between male (47.2%) and female (52.8%), and were mostly married (90.4%). Most of the participants attended secondary school (63.5%) as their highest education and had a mean (SD) monthly income of RM 3575.70 (1189.20). In a further analysis, the mean monthly salary was categorised into two groups using the Malaysian B40 household income as a cut-off point; whereby 348 (65.4%) participants were categorised as low income while 184 (34.6%) were not. For QOL, 143 (27.1%) participants had generally poor health, while 384 (72.9%) had good health. A total of 180 (34.2%) were dissatisfied with their health, while 347 (65.8%) were satisfied. Out of a score of 100, the participants had a mean (SD) physical health of 70.9 (12.2), mean (SD) psychological health of 71.5 (11.8), mean (SD) healthy social relationship of 74.4 (14.6), and mean (SD) environment of 68.5 (12.0).

### 3.2. Association between the Independent Variables and QOL Domains

The logistic regression found a statistically significant association between general QOL and four of the independent variables (*p* < 0.05), as seen in Table 2. These variables were included in the model, along with another eight variables that showed an association with the general QOL at a significance level of less than 0.25. The Hosmer–Lemeshow test (*p* = 0.707), Pearson chi-square (*p* < 0.001), and classification table (overall correctly classified percentage = 75.7) were applied to check the model fitness of final model. The result of the multivariable logistic regression analysis (Table 2) shows that only owning a house outright (AOR 2.55, 95% CI 1.26, 5.18), sufficient health literacy (AOR 4.32, 95% CI 1.93, 9.67), excellent health literacy (AOR 5.92, 95% CI 1.92, 18.23), high financial retirement confidence (AOR 5.71, 95% CI 1.98, 16.49), and one or more extra sources of income (AOR 0.45, 95% CI 0.26, 0.77) were significantly associated with a good QOL when adjusted for other variables in the model. Thus, these five predictors can distinguish people who have a good general QOL from those who do not, where χ^2^ (5, *n* = 453) = 67.033, *p* < 0.001. The model can explain between 13.8% for Cox and Snell R^2^ and 20.0% for Nagelkerke’s R^2^ of the variance of having a good quality of life. The overall prediction success overall is 75.7% of cases.

The same logistical analysis also predicted a statistically significant association between health satisfaction and four of the independent variables (*p* < 0.05) along with seven other variables (*p* < 0.25). The Hosmer–Lemeshow test (*p* = 0.533), Pearson’s chi-square (*p* < 0.001), and classification table (overall correctly classified percentage = 69.2) were applied to check the model fitness of the final model. The result of the multivariable logistic regression analysis shows that only sufficient health literacy (AOR 2.68, 95% CI 1.25, 5.76), excellent health literacy (AOR 6.71, 95% CI 2.20, 20.49), more than two chronic diseases (AOR 0.48, 95% CI 0.30, 0.76), consumption of fruit and vegetables (AOR 4.89, 95% CI 1.05, 22.88), and high financial retirement confidence (AOR 3.11, 95% CI 1.17, 8.27) were significantly associated with health satisfaction when adjusted for other variables in the model. These five predictors were able to distinguish people who were satisfied with their health from those who were not, where χ^2^ (9, *n* = 451) = 56.403, *p* < 0.001. The model is able to explain between 11.9% (Cox and Snell R^2^) and 16.3% (Nagelkerke’s R^2^) of the variance of health satisfaction. The overall prediction success is 69.2% of the cases.

The result of the univariable analysis of the association between each independent variable and the physical health domain shows that eight variables were significantly associated (*p* < 0.05) along with two other variables (*p* < 0.25). The coefficient of determination (adjusted R^2^) = 0.015, intercept = 56.52, R = 0.415, R^2^ = 0.172 and adjusted R^2^ = 0.159. It was noticed that the ‘moderate financial retirement confidence’ and ‘high financial retirement confidence’ variables had multicollinearity with a variance inflation factor (VIF) ranging between 5 and 10. Therefore, ‘moderate financial retirement confidence’, the one with less significant value, was eliminated. In the final model, only eight predictors were significantly associated with physical health. The R for the regression was significant, F (8, 423) = 11.272, *p* < 0.001, with R^2^ = 0.176. The adjusted R^2^ of 0.160 indicates that 16% of the variance in the physical health domain can be predicted by these 8 predictors.

The result of the univariable analysis of the association between each independent variable and the psychological domain identified that six variables are significantly associated (*p* < 0.05) with an additional eight other variables (*p* < 0.25). Therefore, the variable in each pair with the less significant value was eliminated from the model. They were moderate financial retirement confidence and the secondary education variables. In the final model, only seven predictors are statistically associated with the quality of life in the psychological domain. The R for regression is significant, F (7, 432) = 14.415, *p* < 0.001, with R^2^ = 0.189. The adjusted R^2^ of 0.176 indicates that almost 18% of the variance in the psychological domain can be predicted by the 7 predictors.

The univariable analysis of the association between each independent variable and social relationships shows five variables that are significantly associated (*p* < 0.05) with an additional eight variables (*p* < 0.25). Therefore, moderate financial retirement confidence with less significant value was eliminated from the model. In the final model, only six predictors were statistically associated with social relationships. The R for the regression is significant, F (6, 423) = 7.804, *p* < 0.001, with R^2^ = 0.142. The adjusted R^2^ of 0.124 indicates that almost 12.0% of the variance in the social relationship domain can be predicted by those 6 predictors.

The final domain, environment, shows significant association between eight independent variables (*p* < 0.05) with an additional four variables (*p* < 0.25). One pair of variables consisting of ‘moderate financial retirement confidence’ and ‘high financial retirement confidence’ had a VIF between 5 and 10, which indicated that they had multicollinearity. Therefore, the one with the less significant value, which is moderate financial retirement confidence, was eliminated from the model. In the final model, only seven predictors were statistically associated with the environment domain. The R for the regression is significant, F (7, 429) = 21.359, *p* < 0.001, with R^2^ = 0.246. The adjusted R^2^ of 0.246 indicates that almost 25% of the variance in the environment domain can be predicted by these 7 predictors.

### 3.3. Association between Active Ageing Awareness and QOL Domains

In this study, it was hypothesised that higher awareness of active ageing would be associated with a higher QOL. In Table 2 and Table 3, it was reported that awareness of active ageing was not associated with general QOL and health satisfaction but was associated with QOL in all four domains after controlling for other variables. However, it was also noted that awareness of active ageing was not the sole predictor for predicting the QOL in the four domains. There were other statistically significant predictors too. Therefore, hierarchical multiple regression was performed to observe the effect of the awareness of active ageing as one of the predictors on the QOL in the four domains.

In the hierarchical multiple regression analysis, the two confounders (age and educational level) and the other significant predictors for each outcome that were identified during the multivariable regression analysis were entered into an initial model (Model 1). Then, all these variables and the awareness of active ageing were inserted together into Model 2, and the value of the R^2^ change in the two models was observed. Additionally, the standardised regression coefficient of awareness of active ageing was compared with that of the other significant predictors in Model 2. If the value of the standardised regression coefficient of awareness of active ageing is the largest in Model 2, it can be assumed that the awareness of active ageing has the highest significant effect on the outcome as compared to the other predictors.

As seen in Table 4, the physical health domain explains 17.8% of the variance. The variables in Model 1 consist of age, educational level, job category, having a financial loan commitment, health literacy, number of chronic diseases, attended a non-job-related course in the past six months, and financial retirement confidence. However, when active ageing awareness is entered into Model 2, the total variance explained becomes 19.8%, F (16,458) = 7.064, *p* < 0.001. Thus, awareness of active ageing explains 2.0% of the variance in physical health after controlling for other variables, where R^2^ change = 0.020, F change (1458) = 11.427, *p* < 0.001. In the final model for physical health, awareness of active ageing is statistically significant (β = 0.154), but it does not have the highest significant effect. The variable with the highest significant effect in predicting physical health is the number of chronic diseases (no disease vs. more than two diseases, β = 0.254).

In the psychological domain, Model 1 explains 16.3% of the variance. The variables in Model 1 are age, educational level, having other sources of income, health literacy, consumption of fruit and vegetables, attended a non-job-related course in the past six months, and financial retirement confidence. When awareness of active ageing is entered into Model 2, the total variance explained becomes 19.8%, F (13,435) = 8.262, *p* < 0.001. Hence, awareness of active ageing explains about 3.5% of the variance in the psychological domain after controlling for other variables, with R^2^ change = 0.035, F change (1435) = 18.861, *p* < 0.001. The awareness of active ageing is statistically significant (β = 0.196) in predicting the QOL in the psychological domain, but health literacy (excellent vs. inadequate health literacy, β = 0.229) has the highest significant effect.

As for predicting the QOL in the social relationship domain, Model 1 explains 11.6% of the variance. The variables in Model 1 are age, educational level, home ownership, health literacy, attended a non-job-related course in the past six months, and financial retirement confidence. When awareness of active ageing is entered into Model 2, the total variance explained increases to 14.3%, F (14,455) = 5.435, *p* < 0.001. Thus, awareness of active ageing explains 2.7% of the variance on after controlling for other variables, R^2^ change = 0.027, F change (1455) = 14.537, *p* < 0.001. The awareness of active ageing is statistically significant and has the highest significant effect (β = 0.175) in predicting the QOL in the social relationship domain.

Finally, Model 1 explains 24.5% of the variance in the QOL in the environmental domain. The variables in Model 1 are age, educational level, health literacy, consumption of fruit and vegetables, smoking status, attended a non-job-related course in the past six months, and financial retirement confidence. In Model 2, awareness of active ageing increases the total variance explained to 26.3%, F (14,422) = 10.782, *p* < 0.001. In other words, awareness of active ageing explains about 2.0% of the variance after controlling for other variables, R^2^ change = 0.019, F change (1422) = 10.829, *p* < 0.001. In the final model, awareness of active ageing is statistically significant (β = 0.145), but it does not have the highest significant effect. The highest significant effect in predicting QOL in the environment domain is produced by health literacy (excellent vs. inadequate health literacy, β = 0.286).

## 4. Discussion

### 4.1. Association between Active Ageing Awareness and QOL Domains

The present study aimed to assess the factors associated with QOL and its association with active ageing awareness. Multivariable logistic regression analyses were conducted to find the predictors of general QOL and health satisfaction. The results revealed that the significant predictors of general QOL were home ownership, no other sources of income, sufficient and excellent health literacy, and high financial retirement confidence. As for health satisfaction, the significant predictors were found to be not having any chronic diseases, healthy diet, sufficient and excellent health literacy, and high financial retirement confidence.

Multivariable linear regression analyses were then conducted to predict the QOL in each of its four domains. The results showed that high financial retirement confidence, sufficient and excellent health literacy, and awareness of active ageing were positively associated with the QOL in all four domains. Financial retirement confidence is related to financial security in later life as it indicates that the individual is prepared to save enough money for their later life requirements. Health literacy is associated with a good long-term health outcome, [27] and the ability to maintain good health enables the older person to remain active in terms of social participation and being involved in leisure activities as well as physical activities. Thus, health literacy may be associated with the QOL in all four domains.

Furthermore, having a higher awareness of active ageing implies that the individual will have a higher QOL in all four domains. The findings can be explained using the stages of behavioural change model by Prochaska et al., whereby the behavioural change starts once the individual is aware of the problem (Contemplation phase) he or she is going to face in the future, followed by the action to practice the desired behaviour [29] (Action phase). In this study, those who are aware of various ageing issues that they will face in the future may have acted towards it, which was reflected in having higher QOL.

In addition, the physical health domain was significantly associated with lower job category, not having any financial loan commitments, not having any chronic diseases, and having attended a non-job-related course in the past six months. The psychological domain was also associated with the above three predictors, and it was also associated with some other predictors, namely, having other sources of income, having a healthy diet (fruit and vegetable consumption in line with the WHO recommendation), and having attended a non-job-related course in the past six months. As for the social relationship domain, in addition to the awareness of active ageing, health literacy, and financial retirement confidence, it was also associated with educational level, home ownership, and attended a non-job-related course in the past six months. While for the environment domain, the additional predictors were fruit and vegetable consumption and smoking status. All the multivariable analyses were controlled for identified confounders.

The analyses showed that health literacy and financial retirement confidence were significant predictors in ensuring the QOL of future older persons. This finding supports previous studies that found that health literacy is associated with health outcomes [10,30]. A study on retirement confidence in the USA found that being financially prepared for retirement was associated with higher confidence to retire among workers and retirees aged more than 25 years old [31], which may have an influence on an individual’s QOL. On the other hand, those who had to invest their money or do another job to get more income reported having a lower QOL. The motivation to gain extra income may arise from the need to meet household requirements, and thus, the pressure of doing so may affect their QOL. A healthy diet, which was measured by the consumption of fruit and vegetables based on the WHO recommendation, was found to be associated with several outcomes such as general health status and health-related QOL, physical function, and frailty, as well as mental health [32].

### 4.2. Association between Awareness of Active Ageing and Quality of Life

The association between awareness of active ageing and QOL is not well established. In this study, awareness of active ageing was found to have a positive effect on the QOL in the four domains. Moreover, awareness of active ageing was the strongest predictor for QOL in the social relationship domain. Overall, the standardised regression coefficient for awareness of active ageing ranged between 0.145 and 0.196, and the effect size is small. This study can be considered among the first to be conducted on this issue in Malaysia.

In general, awareness studies in Malaysia have found that the awareness level of certain conditions is good. For instance, a study on eye donation found that even though awareness about eye donation is high, the willingness to donate an eye is very poor [33]. Another study on mammography screening found that almost half of Malaysian women are aware of mammography screening; however, the mammography uptake is very low [34]. The association between the awareness and the study outcome in both above-mentioned studies were not significant. These two studies are related to behaviour, willingness to donate eyes and to go for mammography screening. In both studies, the prevalence of the outcomes is also low.

Furthermore, a study on NCDs in Malaysia found that awareness of diabetes mellitus, hypertension, and hypercholesterolemia is associated with having a family history of the disease [35]. This implies that those with exposure to a specific disease or condition have a better awareness. However, the study lacked information on the relationship between awareness of these conditions and behavioural change. From the above examples, it can be concluded that the level of awareness of a specific condition or situation is dependent on the outcome being measured.

### 4.3. Strengths and Limitations of Study

The psychometric evaluation was conducted among employees in the public sector in Malaysia, which involved the use of a moderate sample size and was randomly sampled, which allows strong representation among the study population. This study is also among the first to focus on the QOL of future older people in Malaysia. The results will enable policymakers to estimate the ability of their older population to age actively and to identify the factors that may assist them in enhancing their QOL [36]. Finally, this study opens the opportunity for a future longitudinal study to assess the changes in the QOL of the participants when they are more than 60 years old.

The limitation of this study is that the result is based on the use of a self-report measurement, which is a type of measurement that is prone to response and information bias. In addition, two variables (total value of loans and monthly loan commitment) affected the missing value analysis and were subsequently excluded. The participants were reluctant to answer these questions as they would disclose sensitive information about their wealth. As this is a cross-sectional study design, the causal relationship between the independent variable and the dependent variable cannot be confirmed. Furthermore, recent scientific results indicate a significant role of psychosocial stress in the intensification of ageing processes [37] but were not included in this study. Another limitation is that it involved only employees from the public sector. Nevertheless, even though the findings from this study cannot be inferred to the whole employee population in Malaysia, at least it can be generalised to the non-professional group of public employees throughout the country.

## 5. Conclusions

Ageing is an unavoidable process that will be faced by everyone. In the present study, awareness of active ageing was found to have a positive effect on the QOL in the four domains. Moreover, awareness of active ageing was the strongest predictor for QOL in the social relationship domain. Improvements in the study design can be made using the face-to-face interview technique. This approach may reduce the amount of missing data, particularly in relation to financial information. Regardless, this study can be extended by developing it into a longitudinal study to follow up the participants at five-year intervals until they become older persons and thereby evaluate their QOL in the future. This will provide an evidence base for the ageing process faced by the Malaysian population. A collaboration of multiple stakeholders and incorporation of the programmes using a holistic approach based on the three pillars of the active ageing framework in promotion and intervention for active ageing will maximise the resources in terms of manpower, time, and funds, of every stakeholder. Now is the best time to highlight and bring up the active ageing promotion and intervention programme as a priority of the national agenda, particularly among the B40 population.

## Figures and Tables

**Table 1 ijerph-19-09034-t001:** Socio-demographic and Health-related Characteristics of the Participants (N = 532).

Characteristics		Prevalence, *n* (%)
**Age**		50.2 (5.9) ^a^
40 to 44		147 (27.6)
45 to 49		109 (20.5)
50 to 54		130 (24.4)
55 to 60		146 (27.5)
**Gender**		
Male		251 (47.2)
Female		281 (52.8)
**Ethnicity**		
Malay		512 (96.2)
Chinese		3 (0.6)
Indian		15 (2.8)
Others		2 (0.4)
**Marital Status**		
Never married		18 (3.4)
Married		472 (90.4)
Separated/Divorced		16 (3.1)
Widow/Widower		16 (3.1)
**Highest educational Level**		
Primary school		14 (2.7)
Secondary school		334 (63.5)
Pre-university/Diploma		143 (27.2)
Bachelor’s degree and above		35 (6.6)
**Occupational Category**		
Technicians and associates professional		185 (35.2)
Clerical workers		163 (31.0)
Services and general workers		177 (33.8)
**Monthly Salary**		3575.70 (1189.20) ^a^
<RM 3900 (low-income, B40)		348 (65.4)
≥RM 3900 (Moderate income, non-B40)		184 (34.6)
**Number of Other Sources of Income**		
No other income		157 (29.5)
At least one other income		375 (70.5)
**Home Ownership**		
No house		115 (21.8)
Owns a house, still paying the loan		288 (54.7)
Owns a house (no debt/inherited property)		124 (23.5)
**Number of Loan Commitments**		
No loan		60 (11.3)
At least one		472 (88.7)
**Financial Retirement Confidence**		
Low		24 (4.6)
Moderate		327 (62.9)
High		169 (32.5)
**Health Literacy**		
Inadequate		41 (7.9)
Problematic		182 (35.1)
Sufficient		237 (45.8)
Excellent		58 (11.2)
**Number of Chronic Diseases**		
No disease		270 (50.8)
1 disease only		168 (31.5)
2 or more diseases		94 (17.7)
**Fruit/Vegetable Intake (WHO Recommendation)**		
<5 servings per day		455 (94.8)
≥5 servings per day		25 (5.2)
**Smoking Status**		
Never smoked		424 (80.7)
Ex-smoker		16 (3.0)
Current smoker		86 (16.3)
**Physical Activity (WHO Recommendation)**		
<600 METs minutes per week		133 (25.0)
≥600 METs minutes per week		399 (75.0)
**Awareness of Active Ageing**		71.4 (19.1) ^b^
		532 (100)
**Attended a Pre-Retirement Course**		
No		489 (91.9)
Yes		43 (8.1)
**Attended a Non-Job-Related Course**		
No		441 (82.9)
Yes		91 (17.1)
**Owns a Vehicle**		
No		27 (5.1)
Yes		504 (94.9)
**Has a Valid Driving Licence**		
No		29 (5.5)
Yes		501 (94.5)
**Number of Facilities within 5 Min Walk**		
No facility near home		220 (42.1)
At least one facility near home		302 (57.9)
**Quality of Life (QOL)**		
General health	Poor QOL	143 (27.1)
	Good QOL	384 (72.9)
Health satisfaction	Dissatisfied	180 (34.2)
	Satisfied	347 (65.8)
**Physical Health Domain**		70.9 (12.2) ^a^
**Psychological Domain**		71.5 (11.8) ^a^
**Social Relationships Domain**		74.4 (14.6) ^a^
**Environment Domain**		68.5 (12.0) ^a^

^a^ Mean (SD), ^b^ Median (IQR).

**Table 2 ijerph-19-09034-t002:** The Association between the Independent Variables and QOL Domains.

Independent Variable	Overall, *n* (%)	General QOL		Health Satisfaction		Physical Health		Psychological Health		Social Relationship		Environment	
		Good QOL, *n* (%)/Mean (SD)	Crude Association (*n* = 527) OR (95% CI)	Satisfied with Health, *n* (%)/Mean (SD)	Crude Association (*n* = 527) OR (95% CI)	Mean (SD)	Crude Association Coefficient (*n* = 517) (95% CI)	Mean (SD)	Crude Association Coefficient (*n* = 513) (95% CI)	Mean (SD)	Crude Association Coefficient (*n* = 504) (95% CI)	Mean (SD)	Crude Association Coefficient (*n* = 532) (95% CI)
**Age group**													
40 to 44	147 (27.9)	102 (26.6)	Ref ^e^	98 (28.2)	Ref ^e^	71.2 (12.3)	Ref ^e^	72.6 (11.5)	Ref ^e^	75.2 (14.6)	Ref	68.8 (12.1)	Ref
45 to 49	108 (20.5)	75 (19.5)	1.00 (0.59,1.72)	68 (19.6)	0.85 (0.51,1.43)	70.3 (13.6)	−0.97 (−4.05,2.11)	71.2 (12.6)	−1.39 (−4.37,1.59)	76.0 (13.7)	0.73 (−2.99,4.45)	69.1 (12.4)	0.26 (−2.82,3.34)
50 to 54	127 (24.1)	94 (24.5)	1.26 (0.74,1.13)	83 (23.9)	0.94 (0.57,1.56)	71.9 (11.3)	0.71 (−2.21,3.63)	72.1 (11.4)	−0.50 (−3.33,2.33)	74.3 (13.5)	−0.91 (−4.43,2.62)	69.3 (12.5)	0.43 (−2.48,3.33)
55 to 60	145 (27.5)	113 (29.4)	1.56 (0.92,2.64)	98 (28.2)	1.04 (0.64,1.70)	70.1 (12.2)	−1.17 (−4.01,1.66)	70.1 (11.7)	−2.49 (−5.22,0.24)	75.0 (8.3) ^d^	−2.73 (−6.18,0.71)	67.2 (11.3)	−1.67 (−4.48,1.14)
**Sex**													
Male	250 (47.4)	173 (45.1)	Ref ^e^	169 (48.7)	Ref	71.5 (11.9)	Ref	70.5 (11.6)	Ref ^e^	73.8 (14.8)	Ref	66.6 (11.6)	Ref
Female	277 (52.6)	211 (54.9)	1.42 (0.97,2.09)	178 (51.3)	0.86 (0.60,1.24)	70.3 (12.5)	−1.19 (−3.31,0.92)	72.4 (11.9)	1.83 (−0.21,3.87)	75.0 (14.4)	1.19 (−1.34,3.74)	70.3 (12.2)	3.68 (1.59,5.76)
**Ethnicity**													
Non-Malay	20 (3.8)	16 (4.2)	Ref	14 (4.0)	Ref	73.0 (10.8)	Ref	73.8 (12.4)	Ref	75.0 (16.7) ^d^	Ref	73.3 (13.4)	Ref
Malay	507 (96.2)	368 (95.8)	0.66 (0.22,2.01)	333 (96.0)	0.82 (0.31,2.17)	70.8 (12.3)	−2.23 (−7.70,3.24)	71.4 (11.8)	−2.34 (−7.61,2.94)	74.3 (14.6)	−3.79 (−10.49,2.90)	68.3 (12.0)	−4.94 (−10.33,0.44)
**Marital status**													
No partner	50 (9.7)	36 (9.6)	Ref	30 (8.8)	Ref	69.4 (12.6)	Ref	70.8 (17.7) ^d^	Ref ^e^	70.3 (18.2)	Ref	69.1 (12.3)	Ref
Has partner	467 (90.3)	339 (90.4)	1.03 (0.54,1.97)	309 (91.2)	1.30 (0.72,2.37)	71.1 (12.1)	1.73 (−1.84,5.29)	70.8 (16.7) ^d^	2.57 (−0.88,6.03)	74.5 (14.4)	4.36 (−0.89,9.61)	68.4 (12.1)	−0.71 (−4.29,2.87)
**Education**													
Primary	12 (2.3)	10 (2.6)	Ref ^e^	7 (2.0)	Ref ^e^	73.8 (16.5)	Ref ^e^	71.9 (14.4)	Ref ^e^	69.9 (23.7)	Ref	65.9 (14.5)	Ref
Secondary	332 (63.7)	242 (63.9)	0.54 (0.12,2.50)	224 (65.5)	1.48 (0.46,4.78)	71.0 (11.6)	−2.81 (−9.84,4.22)	71.6 (11.9)	−0.28 (−7.11,6.55)	75.5 (13.7)	5.66 (−2.46,13.77)	68.5 (11.8)	2.63 (−4.11,9.36)
Tertiary	177 (34.0)	127 (33.5)	0.51 (0.11,2.40)	111 (32.5)	1.20 (0.37,3.94)	70.5 (12.9)	−3.28 (−10.42,3.86)	71.1 (11.4)	−0.73 (−7.66,6.20)	75.0 (8.3) ^d^	2.97 (−5.28,11.22)	68.9 (12.5)	2.99 (−3.85,9.83)
**Job category**													
Services	175 (33.7)	128 (33.9)	Ref	117 (34.2)	Ref	71.5 (11.8)	Ref ^e^	71.3 (12.1)	Ref	75.1 (15.0)	Ref	67.3 (12.5)	Ref
Clerical	162 (31.2)	112 (29.6)	0.82 (0.51,1.32)	96 (28.1)	0.72 (0.46,1.12)	69.6 (13.2)	−1.78 (−4.40,0.84)	71.5 (112.5)	0.12 (−2.42,2.66)	75.0 (8.3) ^d^	−1.38 (−4.56,1.81)	68.9 (12.6)	1.59 (−1.02,4.21)
Professionals	183 (35.2)	138 (36.5)	1.13 (0.70,1.81)	129 (37.7)	1.18 (0.76,1.85)	71.5 (11.6)	0.06 (−2.47,2.59)	71.6 (11.0)	0.24 (−2.21,2.70)	75.0 (8.3) ^d^	−0.33 (−3.38,2.73)	69.5 (11.2)	2.20 (−0.33,4.71)
**Income group**													
<RM3900	344 (65.3)	247 (64.3)	Ref	221 (63.7)	Ref	70.8 (12.2)	Ref	71.8 (12.2)	Ref	75.2 (14.7)	Ref	68.4 (12.5)	Ref
≥RM3900	183 (34.7)	137 (35.7)	1.17 (0.78,1.76)	126 (36.3)	1.23 (0.84,1.80)	71.0 (12.2)	0.16 (−2.06,2.37)	71.0 (10.9)	−0.78 (−2.92,1.36)	75.0 (8.3) ^d^	−2.30 (−4.97,0.37)	68.9 (11.3)	0.50 (−1.71,2.70)
**Other income**													
None	155 (29.4)	125 (32.6)	Ref ^e^	102 (29.4)	Ref	71.1 (12.1)	Ref	75.0 (16.7) ^d^	Ref ^e^	75.6 (15.1)	Ref	68.5 (12.9)	Ref
At least one	372 (70.6)	259 (67.4)	0.55 (0.35,0.87) *	245 (70.6)	1.00 (0.68,1.49)	70.8 (12.3)	−0.26 (−2.58,2.07)	70.8 (62.5) ^d^	−1.62 (−3.87,0.62)	73.9 (14.3)	−1.70 (−4.53,1.13)	68.6 (11.7)	0.06 (−2.28,2.40)
**Home Ownership**													
No home	112 (21.5)	79 (20.8)	Ref ^e^	72 (20.9)	Ref ^e^	71.8 (12.7)	Ref	72.2 (12.5)	Ref	75.0 (8.3) ^d^	Ref	69.7 (13.2)	Ref
Home loan	286 (54.8)	199 (52.5)	0.96 (0.59,1.54)	183 (53.2)	0.99 (0.63,1.56)	70.3 (12.0)	−1.51 (−4.20,1.18)	71.1 (11.4)	−1.10 (−3.71,1.52)	75.0 (8.3) ^d^	−2.49 (−5.76,0.80)	68.2 (11.3)	−1.54 (−4.22,1.14)
Owned home	124 (23.8)	101 (26.6)	1.83 (1.00,3.37)	89 (25.9)	1.41 (0.82,2.45)	71.1 (12.3)	−0.71 (−3.87,2.44)	71.8 (12.2)	−0.43 (−3.49,2.64)	75.3 (16.8)	−0.66 (−4.50,3.18)	68.5 (12.7)	−1.22 (−4.39,1.95)
**Loans**													
None	58 (11.0)	48 (12.5)	Ref ^e^	43 (12.4)	Ref ^e^	74.2 (12.2)	Ref ^e^	72.1 (13.1)	Ref	76.2 (16.3)	Ref	68.4 (13.9)	Ref
At least one	469 (89.0)	336 (87.5)	0.53 (0.26,1.07)	304 (87.6)	0.64 (0.35,1.19)	69.4 (12.2)	−3.78 (−7.13,−0.42)	71.4 (11.6)	−0.63 (−3.85,2.60)	75.0 (8.3) ^d^	−1.97 (−6.00,2.06)	68.5 (11.8)	0.12 (−3.18,3.43)
**FRC ^a^**													
Low	24 (4.6)	12 (3.2)	Ref ^e^	11 (3.2)	Ref ^e^	62.4 (13.2)	Ref ^e^	62.5 (10.6)	Ref ^e^	64.9 (20.6)	Ref	57.1 (11.7)	Ref
Moderate	324 (62.7)	220 (58.5)	2.12 (0.92,4.87)	200 (59.0)	1.91 (0.83,4.39)	69.3 (11.4)	6.93 (2.04,11.82)	70.5 (10.7)	8.00 (3.26,12.75)	75.0 (8.3) ^d^	8.47 (2.39,14.56)	66.8 (11.1)	9.69 (4.73,14.65)
High	169 (32.7)	144 (38.3)	5.76 (2.33,14.25) *	128 (37.8)	3.69 (1.54,8.87) *	75.3 (12.2)	12.95 (7.90,18.0)	74.8 (12.7)	12.27 (7.38,17.17)	75.0 (16.7) ^d^	12.80 (6.53,19.08)	73.4 (12.0)	16.34 (11.22,21.45)
**Health literacy**													
Inadequate	41 (8.0)	22 (5.9)	Ref ^e^	19 (5.6)	Ref ^e^	66.8 (14.1)	Ref ^e^	66.8 (12.0)	Ref ^e^	75.0 (16.7) ^d^	Ref	59.4 (14.1) ^d^	Ref
Problematic	180 (35.0)	109 (29.1)	1.33 (0.67,2.62)	101 (29.9)	1.48 (0.75,2.92)	67.6 (11.6)	0.76 (−3.29,4.80)	68.0 (11.2)	1.21 (−2.67,5.08)	75.0 (8.3) ^d^	−1.04 (−5.94,3.87)	64.1 (10.7)	1.33 (−2.49,5.14)
Sufficient	235 (45.7)	194 (51.9)	4.09 (2.03,8.23) *	169 (50.0)	2.97 (1.51,5.83) *	72.4 (11.7)	5.63 (1.68,9.58)	73.3 (11.1)	6.50 (2.71,10.28)	75.0 (16.7) ^d^	5.28 (0.49,10.06)	70.7 (11.2)	8.04 (4.32,11.76)
Excellent	58 (11.3)	49 (13.1)	4.70 (1.84,12.03) *	49 (14.5)	6.30 (2.47,16.13) *	75.8 (11.4)	8.99 (4.23,13.75)	77.2 (11.4)	10.42 (5.86,15.00)	75.0 (16.7) ^d^	8.96 (3.23,14.70)	76.6 (12.1)	13.89 (9.41,18.36)
**Chronic Diseases**													
≥2 diseases	93 (17.6)	63 (16.4)	Ref ^e^	54 (15.6)	Ref ^e^	66.1 (16.1) ^d^	Ref ^e^	66.7 (16.7) ^d^	Ref ^e^	75.0 (8.3) ^d^	Ref	62.5 (12.6) ^d^	Ref
1 disease	164 (31.1)	120 (31.3)	1.23 (0.75,2.26)	95 (27.4)	0.50 (0.33,0.76) *	71.4 (12.5) ^d^	4.70 (1.70,7.73)	70.8 (16.7) ^d^	3.35 (0.33,6.37)	75.0 (8.3) ^d^	2.84 (−0.94,6.62)	68.2 (11.4)	2.65 (−0.52,5.81)
No diseases	270 (51.2)	201 (52.3)	1.39 (0.83,2.32)	198 (57.1)	0.50 (0.31,0.82) *	75.0 (17.9) ^d^	8.62 (5.82,11.43)	75.0 (12.5) ^d^	5.27 (2.47,8.07)	75.7 (15.3)	4.38 (0.89,7.87)	69.7 (12.6)	4.08 (1.14,7.02)
**Fruits and vegetable**													
<5 servings	455 (94.8)	331 (94.0)	Ref ^e^	297 (92.8)	Ref ^e^	70.7 (12.2)	Ref ^e^	70.8 (16.7)	Ref ^e^	74.2 (14.3)	Ref	68.3 (11.7)	Ref
≥5 servings	25 (5.2)	21 (6.0)	1.97 (0.66,5.84)	23 (97.2)	6.12 (1.42,26.84) *	76.6 (12.1)	5.87 (0.95, 10.79)	79.2 (16.7)	7.89 (3.18,12.61)	81.0 (15.5)	6.81 (0.99,12.62)	78.1 (14.9)	9.78 (4.99,14.58)
**Smoking status**													
Smoker	86 (16.4)	54 (14.1)	Ref ^e^	53 (15.3)	Ref ^e^	71.6 (10.9)	Ref ^e^	69.9 (10.9)	Ref ^e^	74.0 (14.8)	Ref	65.3 (10.8)	Ref
Ex-smoker	16 (3.0)	8 (2.1)	0.59 (0.20,1.73)	7 (2.0)	0.48 (0.17,1.43)	61.4 (14.9)	−10.22 (−16.71,−3.72)	66.1 (10.2)	−3.78 (−10.24,2.68)	71.4 (13.4)	−2.65 (−10.47,5.16)	61.7 (9.9)	−3.65 (−10.23,2.93)
Never smoked	423 (80.6)	321 (82.8)	1.87 (1.14,3.05) *	287 (82.7)	1.31 (0.81,2.12)	71.1 (12.2)	−0.51 (−3.35,2.34)	72.1 (11.9)	2.18 (−0.58,4.94)	74.7 (14.6)	0.70 (−2.74,4.14)	69.5 (12.2)	4.17 (1.32,7.02)
**Physical activity**													
<600 METs	131 (24.9)	98 (25.5)	Ref	80 (23.1)	Ref ^e^	70.1 (13.5)	Ref	71.3 (13.4)	Ref	75.0 (17.2)	Ref	69.5 (13.5)	Ref
≥600 METs	96 (75.1)	286 (74.5)	0.88 (0.56,1.38)	267 (76.9)	1.32 (0.87,1.99)	71.1 (11.8)	1.06 (−1.40,3.52)	71.6 (11.2)	0.31 (−2.07,2.69)	74.2 (13.6)	−0.76 (−3.74,2.22)	68.2 (11.5)	−1.28 (−3.75,1.20)
*AAA* ^b^ *score*	521 (100)	75.6 (13.5)	1.01 (0.99,1.02)	75.2 (13.7)	1.00 (0.99,1.01)	75.2 (13.3)	0.16 (0.08,0.24) ^e^	75.2 (13.3)	0.21 (0.14,0.29) ^e^	75.2 (13.3)	0.22 (0.12,0.31)	75.2 (13.3)	0.22 (0.14,0.30)
**PRC ^c^**													
No	484 (91.8)	352 (91.7)	Ref	319 (91.9)	Ref	71.4 (17.9) ^d^	Ref	70.8 (16.7) ^d^	Ref	74.6 (14.5)	Ref	68.5 (12.0)	Ref
Yes	43 (8.2)	32 (8.3)	1.09 (0.53,2.23)	28 (8.1)	0.97 (0.50,1.86)	71.4 (10.7) ^d^	1.22 (−2.60,5.04)	66.7 (12.4) ^d^	−0.61 (−4.38,3.16)	75.0 (8.3) ^d^	−2.47 (−7.13,2.20)	68.6 (12.5)	0.07 (−3.84,3.97)
**Non-job-related course**													
No	437 (82.9)	311 (81.0)	Ref ^e^	280 (80.7)	Ref ^e^	70.2 (71.4)	Ref ^e^	70.6 (11.8)	Ref ^e^	74.0 (14.7)	Ref	67.5 (11.9)	Ref
Yes	90 (17.1)	73 (19.0)	1.74 (0.99,3.07)	67 (19.3)	1.63 (0.98,2.73)	74.1 (12.9)	3.87 (1.11,6.64)	75.7 (10.9)	5.14 (2.48,7.80)	76.3 (13.9)	2.32 (−1.05,5.69)	73.5 (11.7)	6.07 (3.34,8.80)
**Drive licence**													
No	26 (5.0)	22 (5.8)	Ref ^e^	18 (5.2)	Ref	72.6 (11.0)	Ref	74.5 (12.5)	Ref ^e^	75.0 (22.9) ^d^	Ref	70.9 (12.5)	Ref
Yes	499 (95.0)	360 (94.2)	0.47 (0.16,1.39)	328 (94.8)	0.85 (0.36,2.00)	70.7 (12.3)	−1.83 (−6.74,3.09)	71.3 (11.7)	−3.18 (−7.93,1.57)	74.3 (16.6)	−3.88 (−9.87,2.12)	68.4 (12.0)	−2.55 (−7.32,2.21)
**Basic facilities**													
None	218 (42.1)	154 (40.7)	Ref	138 (40.4)	Ref	70.2 (13.1)	Ref ^e^	75.0 (16.7)	Ref ^e^	74.6 (15.9)	Ref	68.8 (12.7)	Ref
At least one	300 (57.9)	224 (59.3)	1.23 (0.83,1.81)	204 (59.6)	1.23 (0.85,1.78)	71.5 (11.5)	1.32 (−0.83,3.47)	70.8 (16.7)	−1.44 (−3.52,0.64)	75.0 (8.3) ^d^	−0.19 (−2.79,2.41)	68.3 (11.5)	−0.49 (−2.64,1.66)

Significance level: * *p* < 0.05, ^a^ Financial Retirement Confidence, ^b^ Awareness of Active Ageing, ^c^ Pre-retirement course, ^d^ Median (IQR), ^e^ variable with *p* < 0.25 in the univariable regression analysis included in the analysis. Backward stepwise method was applied. OR: Odds ratio, SD: Standard deviation, CI: Confidence interval.

**Table 3 ijerph-19-09034-t003:** Multiple Regression Analysis on the Relationship between QOL and AAAQ.

Independent Variable	General QOL	Health Satisfaction	Physical Health	Psychological Health	Social Relationship	Environment
	Adjusted Association (*n* = 453) AOR (95% CI)	Adjusted Association (*n* = 451) AOR (95% CI)	Adjusted Association (*n* = 479) AOR (95% CI)	Adjusted Association (*n* = 452) AOR (95% CI)	Adjusted Association (*n* = 470) AOR (95% CI)	Adjusted Association (*n* = 441) AOR (95% CI)
** *Age group* **						
40 to 44	Ref	Ref	Ref	Ref	Ref	Ref
45 to 49	1.69 (0.87,3.26)	1.28 (0.69,2.38)	0.61 (−2.45,3.66)	0.07 (−2.89,3.03)	1.05 (−2.73,4.82)	1.70 (−1.24,4.63)
50 to 54	1.29 (0.66,2.51)	0.90 (0.48,1.69)	0.66 (−2.30,2.63)	−0.90 (−3.80,2.01)	−1.06 (−4.68,2.77)	0.76 (−2.13,3.65)
55 to 60	2.05 (0.97,4.35)	1.03 (0.53,2.02)	−0.04 (−3.06,2.99)	−0.95 (−3.90,2.00)	−2.68 (−6.84,1.49)	0.33 (−2.53,3.20)
** *Sex* **		-	-		-	
Male	Ref			Ref		Ref
Female	1.05 (0.58,1.90)			−0.99 (−3.52,1.54)		1.25 (−1.25,3.75)
** *Ethnicity* **	-	-	-	-	-	
Non-Malay						Ref
Malay						−3.59 (−8.86,1.68)
** *Marital Status* **	-	-	-			-
No partner				Ref	Ref	
Has partner				1.83 (−1.68,5.34)	4.53 (−0.78,9.85)	
** *Education Level* **						
Primary	Ref	Ref	Ref	Ref	Ref	Ref
Secondary	0.80 (0.12,5.23)	1.84 (0.38,8.95)	−3.82 (−11.60,3.95)	#	−4.72 (−14.22,4.78)	−3.13 (−10.45,4.19)
Tertiary	0.81 (0.12,5.60)	1.28 (0.25,6.56)	−5.44 (−13.51,2.63)	−1.32 (−3.55,0.90)	−3.50 (−6.28,−0.73) *	−4.84 (−12.41,2.73)
** *Job Category* **	-	-		-	-	
Services			Ref			Ref
Clerical			−2.64 (−4.95,−0.33) *			1.15 (−1.49,3.80)
Professionals			1.59 (−1.15,4.33)			2.85 (0.25,5.44)
** *Income Group* **	-	-	-	-		-
< RM3900					Ref	
≥ RM3900					−1.02 (−3.96,1.93)	
** *Other Income* **		-	-			-
None	Ref			Ref	Ref	
At least one	0.45 (0.26,0.77) **			−2.78 (−5.05,0.52) *	−1.23 (−4.24,1.79)	
** *Home Ownership* **			-	-		-
No home	Ref	Ref			Ref	
Home loan	1.35 (0.77,2.40)	1.15 (0.65,2.01)			−3.52 (−6.27,−0.76) *	
Owned home	2.55 (1.26,5.18) *	1.72 (0.83,3.57)			2.18 (−2.18,6.54)	
** *Loans* **				-	-	-
None	Ref	Ref	Ref			
At least one	0.58 (0.22,1.54)	0.58 (0.25,1.39)	−5.05 (−8.60,−1.50) **			
** *FRC* ^a^ **						
Low	Ref	Ref	Ref	Ref	Ref	Ref
Moderate	2.27 (0.85,6.06)	1.70 (0.68,4.30)	#	#	#	#
High	5.71 (1.98,16.49) **	3.11 (1.17,8.27) *	4.08 (1.75,6.40) **	2.65 (0.41,4.88) *	3.63 (0.73,6.53) *	4.03 (1.79,6.26) ***
** *Health Literacy* **						
Inadequate	Ref	Ref	Ref	Ref	Ref	Ref
Problematic	1.28 (0.59,2.80)	1.51 (0.70,3.29)	−1.11 (−5.4,3.19)	0.88 (−3.21,4.97)	0.79 (−6.06,4.48)	0.53 (−3.44,4.50)
Sufficient	4.26 (1.93,9.42) ***	2.68 (1.25,5.76) *	3.18 (0.90,5.46) **	4.02 (1.84,6.19) ***	5.48 (2.67,8.29) ***	5.61 (3.46,7.76) ***
Excellent	6.42 (2.10,19.64) **	6.71 (2.20,20.49) **	4.88 (1.32,8.43) **	7.29 (3.87,10.71) ***	7.39 (3.04,11.74) **	10.28 (6.90,13.66) ***
** *Chronic Diseases* **						
≥2 diseases	Ref	Ref	Ref	Ref	Ref	Ref
1 disease	1.12 (0.57,2.22)	0.48 (0.30,0.76) **	2.28 (−0.98,5.55)	0.99 (−2.15,4.12)	0.80 (−3.18,4.77)	0.33 (−2.78,3.44)
No diseases	1.30 (0.66,2.55)	0.65 (0.37,1.14)	4.00 (1.88,6.12) ***	1.85 (−1.13,4.83)	1.25 (−2.52,5.02)	0.40 (−2.58,3.39)
** *Fruit and Vegetable* **						
<5 servings	Ref	Ref	Ref	Ref	Ref	Ref
≥5 servings	1.04 (0.31,3.45)	4.89 (1.04,22.88) *	2.81 (−1.96,7.59)	4.94 (0.45,9.42) *	4.06 (−1.77,9.89)	6.74 (2.24,11.24) **
** *Smoking Status* **					-	
Smoker	Ref	Ref	Ref	Ref		Ref
Ex-smoker ^d^	0.69 (0.17,2.81)	0.25 (0.06,1.11)	^d^	−2.51 (−9.28,4.27)		−1.21 (−8.12,5.71)
Never smoked	1.75 (0.86,3.40)	1.04 (0.57,1.87)	−0.99 (−3.99,2.00)	0.44 (−2.80,3.68)		2.65 (0.05,5.25) *
** *Physical Activity* **	-		-	-	-	-
<600 METs		Ref				
≥600 METs		1.60 (0.96,2.69)				
***AAA* ^b^ *score***	-	-	0.14 (0.06,0.22) **	0.17 (0.09,0.25) ***	0.17 (0.07,0.27) ***	0.13 (0.05,0.21) **
***PRC* ^c^**	-	-	-	-	-	-
No						
Yes						
** *Non-Job-Related Course* **						
No	Ref	Ref	Ref	Ref	Ref	Ref
Yes	1.68 (0.87,3.26)	1.65 (0.90,3.04)	3.50 (0.78,6.22) *	5.56 (2.91,8.21) ***	3.37 (−0.15,6.89)	5.39 (2.77,8.00) ***
** *Driving Licence* **		-	-			-
No	Ref			Ref	Ref	
Yes	0.85 (0.26,2.77)			−3.36 (−8.08,1.35)	−3.38 (−9.64,2.89)	
** *Basic Facilities* **	-	-			-	-
None			Ref	Ref		
At least one			0.78 (−1.41,2.97)	−1.84 (−3.96,0.28)		

Significance level: * *p* < 0.05, ** *p* < 0.01, *** *p* < 0.001, ^a^ Financial Retirement Confidence, ^b^ Awareness of Active Ageing*,* ^c^ Pre-retirement course, ^d^ Multivariable outliers deleted before running multivariable analysis, #: Excluded from analysis due to multicollinearity issue, AOR: Adjusted odds ratio, SD: Standard deviation, CI: Confidence interval.

**Table 4 ijerph-19-09034-t004:** The Hierarchical Multiple Regression of AAAQ on QOL in Four Domains after Controlling for Other Variables.

Model of the Outcomes	R^2^	Adjusted R Square	R^2^ Change	F Change Statistic	F Statistic	β ^a^
**Physical Health**						
Model 1	0.178	0.151	0.178	6.623 ***	6.623 ***	
Model 2	0.198	0.170	0.020	11.427 **	7.064 ***	
AAA						0.154 **
No disease vs. more than two diseases ^r^						0.254 ***
**Psychological Health**						
Model 1	0.163	0.140	0.163	7.088 ***	7.088 ***	
Model 2	0.198	0.174	0.035	18.861 ***	8.262 ***	
AAA						0.196 ***
Excellent health literacy vs. inadequate health literacy ^r^						0.229 ***
**Social Relationships**						
Model 1	0.116	0.091	0.116	4.598 ***	4.598***	
Model 2	0.143	0.117	0.027	14.537 ***	5.435***	
AAA						0.175 ***
**Environment**						
Model 1	0.245	0.221	0.245	10.533 ***	10.533 ***	
Model 2	0.263	0.239	0.019	10.829 **	10.782 ***	
**AAA**						0.145 **
**Excellent health literacy vs. inadequate health literacy ^r^**						0.286 ***

Significance level: ** *p* < 0.01, *** *p* < 0.001, ^a^ standardized regression coefficient, ^r^ reference, AAA: Awareness of active ageing.

## Data Availability

The datasets used and/or analysed during the current study are available from the corresponding author upon reasonable request.

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
