# Peer review of "Active Ageing Awareness and Quality of Life among Pre-Elder Malaysian Public Employees"

_ijerph, 2022, doi:10.3390/ijerph19159034_

Round 1

Reviewer 1 Report

While the methodology appears sound, the findings of the study are not very interesting as they do not appear to challenge any existing views. It is quite clear that health literacy and strong financial position contribute to higher quality of life. Nonetheless, the paper is methodologically sound and has cited all the relevant literature. Thus, I don't think a further revision is required and the paper can be accepted as is.

Author Response

Dear Reviewer,

Thank you very much for your comments. We appreciate your kind response.

Kind regards.

Reviewer 2 Report

I was really excited to review this paper. Healthy aging and quality of life are very important topics especially as more people have the gift of living longer.

Abstract

There are some minor changes needed. First please avoid stigmatizing ageist language when referring to older people. According to the World Health Organization, ageism is a global challenge https://www.who.int/news/item/18-03-2021-ageism-is-a-global-challenge-un

Use the past tense when reporting on research that has been completed. I have attached  my PDF with comments throughout  

Introduction

I would like to see definitions of the concepts you are planning to investigate. How are you defining quality of life and active ageing.

You have 4 categories of factors related to quality of life—demographics, socio-economic factors, health related factors, and external factors.  

The health-related factors are often referred to as lifestyle factors in which the person has some choice. Isn’t participating in leisure activities a lifestyle health-related choice?

What are external factors? Why is owning a vehicle an external factor and not a socio-economic factor like housing? Is having a pre-retirement plan an external factor or something that people choose to do?  

I would agree the living conditions in a person’s neighborhood is an external factor as are health and social care systems, sense of community, and government policies.  Paragraph 2 on page 2 starting at line 44 needs to be reorganized to lead your reader to the point of this research—" Thus, to optimise the QOL of older people, it is vital to gain a comprehensive understanding of the current QOL of the adult population and of its determinants.”

Currently, to talk about quality of life, make the statement that Thus, QoL needs to be assessed. Then you provide some examples that seem disconnected and then move to the efficacy of health programs.

In each paragraph, you want to take your reader by the hand and lead them to understand your key point. Owl Purdue says it much better than I https://owl.purdue.edu/owl/general_writing/academic_writing/paragraphs_and_paragraphing/paragraphing.html#:~:text=Aim%20for%20three%20to%20five,longer%20paragraphs%20for%20longer%20papers.

Line 60 Page 2 What is “this framework” Be careful so that your reader understands what "This" is.  Can you make it clear that you are proposing a framework for healthy aging or active aging and exactly what the elements of the framework  are?

Again the paragraph beginning on Line 59 jumps around. Can you reorganize this paragraph to lead your reader to say—"Yes this study needs to be done” and understand why.   

Materials and Methods

Can you order this section so that it is in the order in which the study proceeded? Did you do a power analysis to determine the sample size you needed after you recruited participants or before you began recruiting participants? What was your recruiting strategy? Typically the actually description of participants begins at the outset of the results section.

Ethics—well described

2.1. Participant (Line 78) needs to be plural, you have more than one participants.

Tools

Well described.

Statistical Analysis

Again can you check this section to clearly order the steps in your analysis?

Results

Can you check the tense of your verbs? They all   need to be past tense.

After the sentence, ”In a further analysis, the mean 186 monthly salary was categorised into two groups using the Malaysian B40 household in-187 come as a cut-off point.” Can you provide the proportions of people in these categories?

Line 245. Multicollinearity relates back to your theoretical classification of variables in demographics, economic states, health-related factors, and extended factors. One would expect multicollinearity for those factors measuring the same thing.  

I  have a question about variables. It seems to me that some of these variables are ordinal or scale variables. For example,

Line 257 to 258 “One pair of variables – ‘moderate financial retirement  confidence’ and ‘high financial retirement confidence’ – had a VIF between 5-10, indicating that they had multicollinearity. Therefore, moderate financial retirement confidence 259 with less significant value was eliminated from the model.” Theoretically, are these different variables, or are they degrees of the same variable?  If you are using a theoretical framework "this framework" to identify the potential variables you will have a theoretical basis for the variables you enter into your models. I think the entire regression analysis could be more clearly reported if you used a theoretical framework to identify them. Currently this report seems like throwing spaghetti at the wall and hoping that something sticks.  See https://quantifyinghealth.com/variables-to-include-in-regression/  and also Tabachnick, and Fidell, Using Multivariate Statistics, 7th Edition Darlington and Hayes Regression Analysis and Linear Models: Concepts, Applications, and Implementation

Author Response

1

There are some minor changes needed. First please avoid stigmatizing ageist language when referring to older people. According to the World Health Organization, ageism is a global challenge https://www.who.int/news/item/18-03-2021-ageism-is-a-global-challenge-un

Use the past tense when reporting on research that has been completed. I have attached  my PDF with comments throughout  

Typically authors define the factors they are researching. What is quality of life?

Is this moderate to high income-- use the same style as you used above.

Edited

Changed “burden” to “rise.

Minor changes made according to the PDF file provided.

Added

QOL is defined as an individual’s perception of their position in their life in the context of the culture and value systems in which they live and in relation to their goals, expectations, standards and concerns.

Added

(Moderate income, non-B40)

1 (17)

1 (34)

1 (42)

2 (59-61)

5 (310)

2

Introduction

I would like to see definitions of the concepts you are planning to investigate. How are you defining quality of life and active ageing.

Added

QOL is defined as an individual’s perception of their position in their life in the context of the culture and value systems in which they live and in relation to their goals, expectations, standards and concerns.

Active ageing is defined the process of optimising the opportunity of health, participation and security in order to enhance quality of life as people age

2 (59-61)

1 (36-38)

3

You have 4 categories of factors related to quality of life—demographics, socio-economic factors, health related factors, and external factors.  The health-related factors are often referred to as lifestyle factors in which the person has some choice. Isn’t participating in leisure activities a lifestyle health-related choice?

Edited “participation in leisure activities” as health-related factor

In our study, non-job related course was used as a proxy measure for participation in leisure activity. This is any course attended by the participant in the past 6 month, which is not related to their current job.

Nonetheless, the authors agree that leisure activity can be categorised as a lifestyle factor instead of external factors as participants has some choice.

1 (45)

4

What are external factors? Why is owning a vehicle an external factor and not a socio-economic factor like housing? Is having a pre-retirement plan an external factor or something that people choose to do?  I would agree the living conditions in a person’s neighborhood is an external factor as are health and social care systems, sense of community, and government policies.   population and of its determinants.”

Edited “owning a vehicle” as a socio-economic status

In this study, other factors that cannot be classed as demographic, socio-economic or health-related are grouped into the external factors category.

With that being said, the authors agree that owning a vehicle can be categorised as a socio-economic factor instead of an external factor as it contributes to a person’s socioeconomic status.

1 (43)

5

Paragraph 2 on page 2 starting at line 44 needs to be reorganized to lead your reader to the point of this research—" Thus, to optimise the QOL of older people, it is vital to gain a comprehensive understanding of the current QOL of the adult.

Currently, to talk about quality of life, make the statement that Thus, QoL needs to be assessed. Then you provide some examples that seem disconnected and then move to the efficacy of health programs.

Moved and edited sentence

2 (66-68)

6

In each paragraph, you want to take your reader by the hand and lead them to understand your key point.

Owl Purdue says it much better than I (link) 

Thank you for your kind suggestion and have considered its takeaway points:

·       Put only one main idea per paragraph.

·       Aim for three to five or more sentences per paragraph.

·       Include on each page about two handwritten or three typed paragraphs.

·       Make your paragraphs proportional to your paper. Since paragraphs do less work in short papers, have short paragraphs for short papers and longer paragraphs for longer papers.

·       If you have a few very short paragraphs, think about whether they are really parts of a larger paragraph—and can be combined—or whether you can add details to support each point and thus make each into a more fully developed paragraph.

-

7

Line 60 Page 2 What is “this framework” Be careful so that your reader understands what "This" is.  Can you make it clear that you are proposing a framework for healthy aging or active aging and exactly what the elements of the framework  are?

Edited to included elements of framework (active ageing)

By optimising opportunities of health, participation and security, policymakers will be able to plan early interventions to assist the future older person to age gracefully by enhancing their QOL.

2 (73-74)

8

Again the paragraph beginning on Line 59 jumps around. Can you reorganize this paragraph to lead your reader to say—"Yes this study needs to be done” and understand why.  

Moved and restructured paragraph

2 (80-83)

9

Materials and Methods

Can you order this section so that it is in the order in which the study proceeded? Did you do a power analysis to determine the sample size you needed after you recruited participants or before you began recruiting participants? What was your recruiting strategy?

Typically the actually description of participants begins at the outset of the results section.

Moved sample size explanation from “2.2 Procedure” to “2.1 Participants”

The sample size was calculated before the recruitment.

Expanded explanation

The authors attended the District Action Meeting on November 2016 and acquired a name list of 1559 names from 46 departments. The names were combined into a sampling frame and was assigned a unique number. Based on the sample size requirement, a simple random sampling was done and the selected participants were grouped based on their department.

The selected participants were approached through their department. The representative of the department was asked to handover the questionnaire package in the sealed envelopes to the selected participants, and the researcher then collected them within two weeks.

Moved to results

2 (101-104)

2-3 (106-168)

4 (253-255)

10

Ethics—well described

Thank you.

11

2.1. Participant (Line 78) needs to be plural, you have more than one participants.

Edited

2 (95)

12

Tools - Well described.

Thank you.

13

Statistical Analysis - Again can you check this section to clearly order the steps in your analysis?

Restructured into 3 sections

·       2.6.1. Descriptive Analysis

·       2.6.2. Inferential Analysis

4 (225)

(233)

14

Results

Can you check the tense of your verbs? They all need to be past tense.

Edited

Various

15

After the sentence, ”In a further analysis, the mean 186 monthly salary was categorised into two groups using the Malaysian B40 household in-187 come as a cut-off point.” Can you provide the proportions of people in these categories?

Added

; whereby, 348 (65.4%) participants were categorised as low income while 184 (34.6%) were not.

4 (264-265)

16

Line 245. Multicollinearity relates back to your theoretical classification of variables in demographics, economic states, health-related factors, and extended factors. One would expect multicollinearity for those factors measuring the same thing.

I  have a question about variables. It seems to me that some of these variables are ordinal or scale variables. For example, Line 257 to 258 “One pair of variables – ‘moderate financial retirement  confidence’ and ‘high financial retirement confidence’ – had a VIF between 5-10, indicating that they had multicollinearity.

Therefore, moderate financial retirement confidence 259 with less significant value was eliminated from the model.” Theoretically, are these different variables, or are they degrees of the same variable?  If you are using a theoretical framework "this framework" to identify the potential variables you will have a theoretical basis for the variables you enter into your models. I think the entire regression analysis could be more clearly reported if you used a theoretical framework to identify them. Currently this report seems like throwing spaghetti at the wall and hoping that something

sticks. 

See https://quantifyinghealth.com/variables-to-include-in-regression/  and also Tabachnick, and Fidell, Using Multivariate Statistics, 7th Edition Darlington and Hayes Regression Analysis and Linear Models: Concepts, Applications, and Implementation

The two categories, moderate and high financial retirement confidence belong to the same variable and thus have no multicollinearity. The authors

mistakenly added during the write up and apologize for the mistake and therefore, have deleted that line.

10 (373) and 10 (383)

Reviewer 3 Report

Dear Authors,

Thank you very much for the opportunity to review this interesting paper. I fully agree with the authors that increasing life expectancy has led to a global burden of late-life diseases and that quality of life, as well as satisfaction with life, are important health determinant (https://pubmed.ncbi.nlm.nih.gov/29892774/). Awareness of active ageing was have a positive effect on the QOL in the four domains, especially in the social relationship domain. Therefore, the authors indicated the best space for action in the field of active aging promotion programs and awareness of such aging. 

The theoretical chapters are written with great scientific insight, and the resulting chapters present the authors' extensive experience in this type of research. 

The comments are listed below:

1. Keywords should be different than the words in the title.

2. Many scientific results indicate a significant role of psychosocial stress in the intensification of aging processes (https://pubmed.ncbi.nlm.nih.gov/35564437/). The authors, despite a very well-prepared theoretical introduction, do not mention this factor. Perhaps it is worth considering appropriate additions to the introductory chapter. 

3. Very interesting and professional manuscript.

Thank you for the opportunity to review this article.

Author Response

1

Thank you very much for the opportunity to review this interesting paper. I fully agree with the authors that increasing life expectancy has led to a global burden of late-life diseases and that quality of life, as well as satisfaction with life, are important health determinant (https://pubmed.ncbi.nlm.nih.gov/29892774/). Awareness of active ageing was have a positive effect on the QOL in the four domains, especially in the social relationship domain. Therefore, the authors indicated the best space for action in the field of active aging promotion programs and awareness of such aging. 

The theoretical chapters are written with great scientific insight, and the resulting chapters present the authors' extensive experience in this type of research. 

Thank you for your kind words and appreciate your response.

-

2

Keywords should be different than the words in the title.

Noted with thanks.

1 (31)

3

Many scientific results indicate a significant role of psychosocial stress in the intensification of aging processes (https://pubmed.ncbi.nlm.nih.gov/35564437/). The authors, despite a very well-prepared theoretical introduction, do not mention this factor. Perhaps it is worth considering appropriate additions to the introductory chapter. 

Added

Furthermore, recent scientific results indicate a significant role of psychosocial stress in the intensification of aging processes37 but were not included in this study.

16 (580-582)

4

Very interesting and professional manuscript.

Thank you for the opportunity to review this article.

Thank you for your kind words and appreciate your response.

-

Reviewer 4 Report

This manuscript describes a well-presented cross-sectional study and reports on the findings regarding awareness of active ageing and QOL in the Malaysian population. The methodology and descriptive statistics are presented in detail and in an organized structure. It is also observed the findings of this work have been linked with recent studies and discussed with relevance. The authors have made good efforts to present the limitations of the current study and perspectives on future investigations. The current work is a significant and timely contribution when we look at the worldwide ageing population, with growing research on factors associated with QOL and healthy ageing. I have a few suggestions below for consideration:

1.

Ln 188-189, the numbers 143 and 384 should be referring to 27.1% and 72.9%, respectively. Please revise the percentages presented here. Similarly, in Table 1 regarding the demographic details, corresponding revisions should be made for the pair of data displayed in the table on page 6.

2.

Ln 186-188: after the description of categorization into two groups regarding the monthly salary, it would be worthwhile to add the information that 65.4% of respondents belong to the low-income B40 group.

3.

Section 3.3, for the description on p values, it should be written as p<0.001 instead of p=0.001 (e.g. ln 306, 308, 319, 328, etc.).

4.

Ln 421: it is written “both the public and private sectors in Malaysia”, or the study was only involving the public employees as also mentioned in other parts of the manuscript?

5.

Ln 70: the significance of the current study can be more explicitly stated here, can take reference from the statement made in ln 398 and ln 403.

6.

Ln 74-75: the names given here should be consistent with the committee information given in the Institutional Review Board Statement (ln 474-475).

Some editing issues are also suggested, e.g.

1.

Ln 113: “four and 20” is suggested to be written as “4 and 20”.

2.

Ln 125: should be “…converted into a score based on a scale of 0-100”.

Author Response

1

This manuscript describes a well-presented cross-sectional study and reports on the findings regarding awareness of active ageing and QOL in the Malaysian population. The methodology and descriptive statistics are presented in detail and in an organized structure. It is also observed the findings of this work have been linked with recent studies and discussed with relevance. The authors have made good efforts to present the limitations of the current study and perspectives on future investigations. The current work is a significant and timely contribution when we look at the worldwide ageing population, with growing research on factors associated with QOL and healthy ageing.

Thank you for your kind words and appreciate your response.

-

2

Ln 188-189, the numbers 143 and 384 should be referring to 27.1% and 72.9%, respectively. Please revise the percentages presented here. Similarly, in Table 1 regarding the demographic details, corresponding revisions should be made for the pair of data displayed in the table on page 6.

Edited the percentages in text and Table 1

4 (266)

3

Ln 186-188: after the description of categorization into two groups regarding the monthly salary, it would be worthwhile to add the information that 65.4% of respondents belong to the low-income B40 group.

Added

; whereby, 348 (65.4%) participants were categorised as low income while 184 (34.6%) were not.

4 (264-265)

4

Section 3.3, for the description on p values, it should be written as p<0.001 instead of p=0.001 (e.g. ln 306, 308, 319, 328, etc.).

Edited p=0.001 to p<0.001

Various

5

Ln 421: it is written “both the public and private sectors in Malaysia”, or the study was only involving the public employees as also mentioned in other parts of the manuscript?

Removed “Private”

16 (566)

6

Ln 70: the significance of the current study can be more explicitly stated here, can take reference from the statement made in ln 398 and ln 403.

Added

“This study can be considered among the first to be conducted on this issue in Malaysia.”

2 (83-84)

7

Ln 74-75: the names given here should be consistent with the committee information given in the Institutional Review Board Statement (ln 474-475).

Some editing issues are also suggested, e.g.

Edited

“Ethical approval was obtained from the Ministry of Health, Malaysia (NMRR-16-40-28747), Medical Research & Ethics Committee and Medical Ethics Committee University Malaya Medical Center (MREC ID no: 20161-2037).”

2 (89-91)

8

Ln 113: “four and 20” is suggested to be written as “4 and 20”.

Edited four to 4

3 (183)

9

Ln 125: should be “…converted into a score based on a scale of 0-100”.

Edited

“converted into a score based on a scale of 0-100”

3 (195)